

# Named entity recognition and emotional viewpoint monitoring in online news using artificial intelligence

Manzi Tu

School of Humanities and Communication, Hubei University of Science and Technology, Xianning, Hubei, China

## ABSTRACT

Network news is an important way for netizens to get social information. Massive news information hinders netizens to get key information. Named entity recognition technology under artificial background can realize the classification of place, date and other information in text information. This article combines named entity recognition and deep learning technology. Specifically, the proposed method introduces an automatic annotation approach for Chinese entity triggers and a Named Entity Recognition (NER) model that can achieve high accuracy with a small number of training data sets. The method jointly trains sentence and trigger vectors through a trigger-matching network, utilizing the trigger vectors as attention queries for subsequent sequence annotation models. Furthermore, the proposed method employs entity labels to effectively recognize neologisms in web news, enabling the customization of the set of sensitive words and the number of words within the set to be detected, as well as extending the web news word sentiment lexicon for sentiment observation. Experimental results demonstrate that the proposed model outperforms the traditional BiLSTM-CRF model, achieving superior performance with only a 20% proportional training data set compared to the 40% proportional training data set required by the conventional model. Moreover, the loss function curve shows that my model exhibits better accuracy and faster convergence speed than the compared model. Finally, my model achieves an average accuracy rate of 97.88% in sentiment viewpoint detection.

## INTRODUCTION

Traditional radio and television news media have evolved into the digital era, with communication methods increasingly shifting towards network-based platforms, thus diversifying communication. In recent years, the advent of Internet technology has given rise to many online media platforms, disrupting the traditional modes of news dissemination on radio and TV and dramatically increasing the volume of online news information. The abundance of information eliminates the problem of lack of information, but it also brings another challenge—how users can obtain effective information that meets their needs in the massive data. In this context, online news has emerged as a primary

Corresponding author
Manzi Tu, 18608689092@163.com

source of information for people. However, with the enormous amount of news reports available online, people require technology that can help them quickly identify key topics and hot issues in daily news reports.

By analyzing and studying the data generated internally by various news media departments and organizations and extracting the required key information, it is possible to address, to some extent, the challenges of extracting focus and hot topics in news reports. However, obtaining valuable and structured information directly from disorganized and varied-quality text data is difficult. Therefore, there is an urgent need for targeted text processing technology to convert large amounts of unstructured text into structured information and extract valuable content.

Natural Language Processing (NLP) technology (*Khurana, Koli & Khatter, 2023*) is an integrated interdisciplinary discipline in artificial intelligence, linguistics, and computer science. Named Entity Recognition (NER) (*Liu et al., 2022*), a branch of natural language processing tasks, aims to identify and classify entities in complex texts from external sources, including special nouns and time and date phrases in texts. With continued research and technological development, NER has gradually extended to mining hot topics and detecting emotional views on daily news reports. This information can then be presented to users in various ways, enabling them to quickly stay informed about the latest news and make informed decisions. This approach also facilitates the effective use of network resources and improves information utilization (*Yue et al., 2022*).

Named entity recognition is a basic and important research topic in the field of artificial intelligence and even natural language processing. The continuous updating and progress of named entity recognition technology has important research significance and broad research prospects for the development of related fields. The current research on NER is essentially modeling the serialization of text data. A piece of text data is regarded as a text sequence, and then the sequence model is used to process the text sequence and extract the semantic information. On this basis, text structure, dictionary and other formal features are added for interaction and collaboration as a means of data enhancement to improve NER performance. This approach based on sequence modeling has major defects. The main reason is that the method of modeling the serialization of text is more inclined to think that each word in a piece of text is only strongly related to the word before and the word after it. However, in reality, every word in a text not only has a great relationship with its immediate text, but also may have a relatively great relationship with its distant context. Especially for Chinese, this long-distance semantic connection phenomenon is more obvious.

In recent years, neural network models (*He et al., 2022*) have become increasingly popular for NER tasks due to their exceptional performance. However, training these models requires a significant amount of labeled data, one of the current research's major bottlenecks. Despite this, some supervised deep learning methods, such as BiLSTM-CRF (*Luo et al., 2018*), have performed relatively well in NER tasks under sufficient training data. Nevertheless, such models often require extensive training data to achieve generalization, which can be prohibitively expensive, particularly in online news media. The news content text is comprehensive, and performing annotation requires a certain

level of expertise. Therefore, it is critical to explore ways to combine existing technical methods to achieve high-quality NER with minimal annotated data while also performing content sentiment detection in the current online news field. The author proposes an automatic labeling method for Chinese entity triggers and a NER model for small training datasets to address this issue. The author uses additional supervision of training data to jointly train sentence vectors and trigger vectors through trigger-matching networks, with the trigger vector serving as the attention query for the subsequent sequence annotation model. Furthermore, the author extracts explicit sentiment features from the news and extends the sentiment lexicon of network words. By fully considering these explicit sentiment features, the author can frequently mine the obtained sentiment feature items, enabling us to retain the sentiment features of network information better and achieve sentiment observation. In summary, improving the performance of Chinese NER models using a small amount of annotated data is a highly relevant problem, particularly in some special domains where obtaining a large amount of annotated data is difficult or costly.

## RELATED WORKS

### NER technology

The early NER technology is mainly based on the lexicon of different types of datasets, grammar rules and other linguistic features for classification recognition, as shown in Fig. 1. After a series of operations, entity recognition is primarily performed in vector representation.

With the ongoing development of both traditional machine learning and deep learning technologies and the significant increase in server computing power, the intelligent selection of dataset features and the accuracy of algorithm recognition have been gradually increasing. As a result, the field of NER has witnessed advancements in three directions: fine-grained NER, nested NER, and low-resource named recognition. In recent years, deep neural network-based approaches have substantially improved the effectiveness of natural language processing tasks. The advent of word vector tools has enabled the transformation of words into dense vectors that contain richer semantic information than manually selected features and thus help enhance the performance of downstream tasks. Word2vec (*Ghafari et al., 2019*), GloVe (*Zhu, Sun & Lee, 2022*), and BERT (*Alammary, 2022*) are widely employed in NER tasks and have significantly improved the overall model training efficiency. A neural network-based NER model was introduced that used a fixed-size window for each word but did not consider the association information between long text words (*Sun & Li, 2023*). To address this problem, another study (*Sukardi et al., 2020*) designed a BiLSTM-CNNs model that could automatically extract text and text-level features. This approach was extended in a subsequent study (*Luboshnikov & Makarov, 2021*) to a BiLSTM-CNNs-CRF model that added a conditional random field (CRF) layer to optimize the output label sequences. Another study (*Shi, 2022*) proposed an intelligent task recognition model called LM-LSTM-CRF that could extract character-level vector representations from multiple text structures and thus better identify new entities. In another study (*Kim et al., 2020*), the classical LSTM+CRF structure was employed for NER. In a survey of Chinese NER (*Tang et al., 2022*), a lattice-LSTM structure was

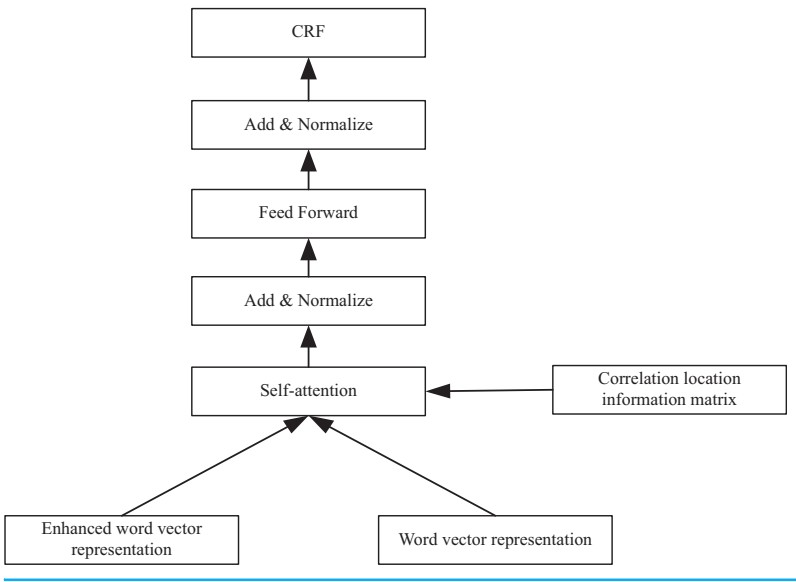

**Figure 1  The implementation of CRF.**     

proposed to mine better the character features in Chinese. Various word segmentation results are introduced into the model, and word information is remotely transmitted to nodes to form a "grid structure" to improve efficiency.

Currently, due to the increasing influence of pre-training models such as the Bidirectional Encoder Representations from Transformers (BERT) in natural language processing (NLP) research, another study (*Chen et al., 2021*) introduced pre-training models in NER research to enhance NER models through the powerful semantic representation ability of pre-training models for semantic understanding of text, thus achieving better entity recognition results.

The advancement of NER models relies heavily on large-scale annotated datasets, where target entities are manually pre-labeled to improve the performance of the trained models. However, this approach poses a challenge in the online news domain due to the scarcity of annotated data. Consequently, several approaches have emerged to address low-resource NER, including NER under additional supervision, NERby self-training methods, and NERby explanatory learning. Among these approaches, NERunder additional supervision, typically involves encoding models based on word embeddings and unsupervised language models, which can incorporate lexical labels, syntactic components, and dependencies. While encoding syntactic information into NER models can compensate for the lack of training data, the traditional approach of encoding linguistic information through word vector representations cannot capture various types of syntactic information. Recent studies, as demonstrated in the literature (*Zhang & Li, 2021*), have proposed using a Key-Value Memory Network (KVMN) to effectively capture supervised information from different sources. KVMN networks have been shown to improve question-answering tasks by encoding contextual information of labels as keys and syntactic information as values. These are subsequently weighted and combined to comprehensively represent the values and connect them to the contextual features.

## Emotional analysis

The predominant techniques for conducting sentiment analysis can be broadly categorized into two distinct methods: semantic-based sentiment analysis and machine learning-based sentiment analysis. The former centers on constructing sentiment lexicons and correlation models between sentiment words to ascertain utterances' positive and negative orientations. In contrast, the latter focuses on leveraging machine learning techniques and vast amounts of data to derive the underlying sentiment orientation of an utterance through statistical inference and implicit mathematical functions. This approach has garnered considerable attention in recent times.

In their work, *Bălan et al. (2019)* employed machine learning methods to perform sentiment classification. However, they opted to utilize only the top-rated words according to a prescribed scoring criterion rather than incorporating all available words. Similarly, *Li, Kang & Sohaib (2023)* applied a support vector machine (SVM) approach to evaluate the relevance of words to the underlying topic and content, with the resulting relevance scores proving invaluable in extracting effective word features. By adopting a bag-of-words model, *Asif et al. (2020)* developed a novel approach to sentiment classification using Twitter data, demonstrating this technique's viability in the context of social networks. Notably, the authors also leveraged syntactic spanning trees of sentences to capture more pertinent inter-word relationships. A maximum entropy approach was deployed to analyze sentence syntax trees, revealing the grammatical rules embedded therein (*Liu et al., 2020*). In *Aljuaid et al. (2021)*, The authors attempted to deduce the sentiment tendencies of sentences by employing both utterance features and structural attributes and establishing manual judgment rules. Finally, *de Souza Amorim & Visser (2020)* analyzed grammar trees through a combination of rules and patterns, significantly enhancing analysis efficiency relative to previous approaches. However, they did not achieve the same proficiency in statistical inference and implicit representation through mathematical functions (*Alharbi & Sohaib, 2021*).

## MODEL DESIGN

In this chapter, the construction of the scheme is expanded. In the named entity is the quilt, the entity trigger is the context in the text that is closely related to the entity. Based on the research into Chinese texts' written expressions and grammar, the author has found that the correlation between Chinese entities and their surrounding words surpasses that of other languages. Consequently, the author has opted to adopt a procedural approach to automatically tag triggers and carry out sentiment detection predicated on successful NER. Next, the author will first define the named entity recognition model in network news, and on the basis of the trigger matching network and the construction of the whole scheme.

### Network news NER model

The model I designed mainly consists of three parts: TrigEncoder, TrigMatcher and SeqTagger. The model structure is depicted in Fig. 2. The Trigger Encoder module is responsible for acquiring the vector representation of entity triggers and conducting

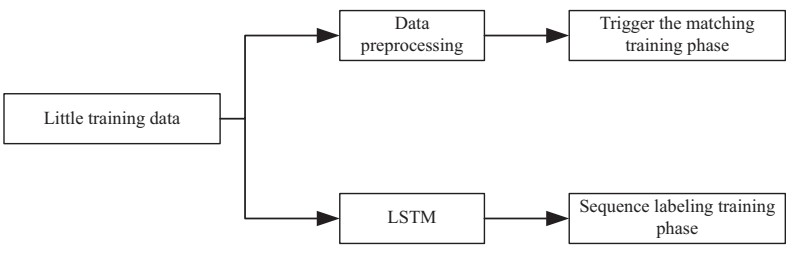

**Figure 2  Model structure diagram.**

multi-classification of triggers based on entity types. The trigger semantic matching module accomplishes semantic matching of training samples and entity triggers, enabling inference of similar entity triggers for new samples. Lastly, by merging prior trigger representations, the NER sequence annotation module conducts sequence annotation on unannotated sentences.

As shown in Fig. 2, in the trigger matching training stage, a small amount of training data is preprocessed to get entity trigger annotation, and then the sentence vector representation and trigger vector representation are obtained through Mogrifier LSTM layer and the self attention layer. Finally, the two are jointly trained to get the optimal representation. The purpose of this stage is to train the trigger vector and get the trigger vector table of the training set. In the sequence annotation training stage, a small amount of training data is encoded by the LSTM layer, the sentence hidden vector is connected with the matching hidden vector, and then sent to the CRF layer for sequence annotation training. The task of this stage is to train the trigger enhanced sequence labeling model. In the test phase, the author get the sentence hidden vector by using the trained LSTM layer, and train the trigger vector that matches it best in the trigger vector table of the training set.

## Mogrifier triggered matching network

The author has developed a Chinese trigger-attention-enhanced sequence representation model, the Mogrifier Trigger Matching Network. This model builds upon efficient Chinese triggers, improves upon the original trigger encoding method, and incorporates the Mogrifier-LSTM structure to address the context-independent representation of input tokens. In the subsequent sections, the author will elaborate on the network structure of each stage of the model training process.

In the initial stage of the modeling framework, the trigger matching network deploys a shared embedding space to jointly train both the trigger encoding module and the attention-based trigger matching module.

Specifically, for a sentence x containing $n$ entities $\{a_1, a_2, ..., a_n\}$, for each entity $a_i (1 \leq i \leq n)$, let there exist a set of triggers

$$T_i = \{t_{i1}, t_{i2}, ...\} \tag{1}$$

It is subsequently classified using the trigger matching network using the entity types corresponding to the triggers as supervision. Thus, the trigger vector $g_t$ is further sent to

the multiclass classifier to predict its corresponding entity type. The loss function for trigger classification:

$$L_1 = -\sum \log P(type(a)|g_t; \theta_{TC}) \tag{2}$$

To accomplish this, the author utilizes the contrast loss function to facilitate learning of the matching between triggers and sentences. During the training process of the matching module, the author randomly shuffles the triggers and sentences, thereby generating two training instances—matching and mismatching—from which the negative samples are required for matching training. The contrast loss function for matching is defined as follows, as shown in Eqs. (8) and (9):

$$d = ||g_s - g_t||_2 \tag{3}$$

$$L_2 = (1 - \beta) \times \frac{1}{2} d^2 + \frac{1}{2}\beta \times [\max(0, m - d)]^2 \tag{4}$$

$$\beta = \begin{cases} 0, \text{mismatch} \\ 1, \text{match} \end{cases} \tag{5}$$

Therefore, the joint loss function of the triggered matching network can be defined as $L = L_1 + \alpha L_2$, where $\alpha$ is the weight to be adjusted. Subsequently, given a sentence x, the author use the previously trained trigger matching network to compute the average of all trigger vector scores associated with this sentence x. Following the traditional attention approach, the author take the average trigger vector $g_t$ as the attention query and create a series of attention-based hidden state labels, denoted as *Tag*:

$$a* = SoftMax(v^T \tanh(U_1 H^T + U_2 g_t^T)^T) \tag{6}$$

$$Tag = a * H \tag{7}$$

where $U_1$, $U_2$ and v are the training parameters for computing the trigger-enhanced attention scores for each hidden state label. Finally, the author concatenates the original hidden state label $H$ with the trigger-enhanced hidden state label $\tilde{H}$ as the input to the final CRF annotator $[H; \tilde{H}]$ to obtain the sentence entity labels for the prediction model.

### Emotional analysis and monitoring

In the case of online news, after the process of NER, the media system tracks the sentiment viewpoints based on sentence entity tags. If single entity tags are used for description, it may result in ambiguity. However, using a multi-vocabulary approach for characterization minimizes the possibility of ambiguity. In the case of a single-vocabulary description of the monitored object, many words related to the object may fall outside the scope of monitoring. However, using a multi-vocabulary description, words similar to all words can be reasonably considered as extended neologisms. This is illustrated in Eq. (8):

$$sim(w, o) = \prod_{i}^{o} sim(w, o_i) \tag{8}$$

It can be considered that the similarity of an observed view $o$ to a vocabulary $w$ is equal to the product of the similarity of all o the entity labels $o_i$ and the vocabulary $w$. After calculating this product, the vocabulary with the highest final similarity is then extracted as a new word extension.

Upon developing an efficient online neologism detection algorithm for news articles, subsequent sentiment topic detection can be carried out. The system user can tailor the sensitive word list and the number of newly-introduced words to be detected, thus catering to their business requirements enabling the system to learn and assimilate fresh vocabulary. Initially, assuming that any monitoring entity or vocabulary must align with the overall paragraph theme is plausible, thus necessitating its prevalence throughout most of the sentences. Based on this premise, it can formulate the following expression.

$$p(w|content) = \sum_{i}^{n} \gamma_i \times p(w|x_i) \tag{9}$$

where $p(w|content)$ denotes the probability that viewpoint is w in the passage, and $p(w|x_i)$ denotes the probability that viewpoint is w is reasonable in the contextual setting of the sentence $x_i$, i.e., the probability that viewpoint w can be the topic of discussion in the sentence $x_i$. $\gamma_i$ is a weighting additive to $p(w|x_i)$ that is based on the distance of the sentence from viewpoint w.

Utilizing Eq. (10) as a foundation, the author devised a neural network architecture for monitoring sentiment analysis of online news, incorporating five input parameters in the input layer. These parameters essentially match the target vocabulary and its five closest sentences, as presented in Fig. 3. The hidden layer is composed of four nodes, with the input for each layer of neurons being defined as follows:

$$input = \sum x_i w_i \tag{10}$$

where $x_i$ is the output value of each input element and $w_i$ is the weight on its input path. Then the output of each layer of neurons is obtained by Sigmoid function.

$$output = sig\text{moid}(\text{input}) = \frac{1}{1 + e^{-\text{input}}} \tag{11}$$

The model is trained by iteratively computing the residuals through each layer in a backward-to-forward fashion and updating the weight values accordingly using a gradient descent approach. Through exposure to a substantial volume of training data, the neural network can conduct nonlinear weight computation and analysis, enhancing the precision of sentiment view detection.

## EXPERIMENTS AND RESULTS

To further evaluate the effectiveness of the proposed model, as shown in Table 1, the author used Inteli7-8700, NVIDIA GeForce RTX 2070, under 16 G RAM, using development language Python3.7 and deep learning framework Pytorch1.14, under Win10 OS. Next, the author will first define the named entity recognition model in network news,

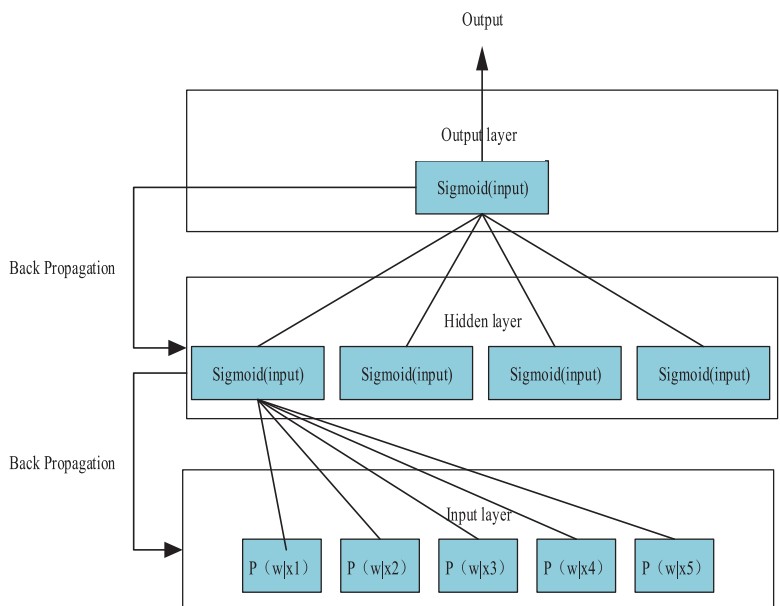

**Figure 3 Neural network algorithm for affective opinion monitoring.** These parameters are essentially the matches between the target vocabulary and its five closest sentences.

**Table 1 System parameter.**

| Index | Argument |
| --- | --- |
| Processor | Inteli7-8700, NVIDIA GeForce RTX 2070 |
| Development language | 16G |
| Frame | Pytorch1.14 |
| Operating system | Win10 |

and on the basis of the trigger matching network and the construction of the whole scheme.

## Data processing

To demonstrate the efficacy of the sequence labeling model proposed in this article, two evaluation metrics—Precision (P) and F1-score—are employed to compare its performance in dichotomous classification tasks. The comparison of true values and predicted results of the model is classified into four categories: true positives (TP), true negatives (TN), false positives (FP), and false negatives (FN). Before experimenting, a dataset of online news domains is required, which is obtained by utilizing the ResumeNER *corpus* to identify the named entities of financial domain information in online news, thereby enabling sentiment monitoring based on it. Subsequently, training data proportions ranging from 20% to 50% are selected from the experimental *corpus* as training sets for comparison scheme models such as BiLSTM-CRF (*Luo et al., 2018*), LM-LSTM-CRF (*Shi, 2022*), Lite-LSTM (*Tang et al., 2022*), and trigger matching networks

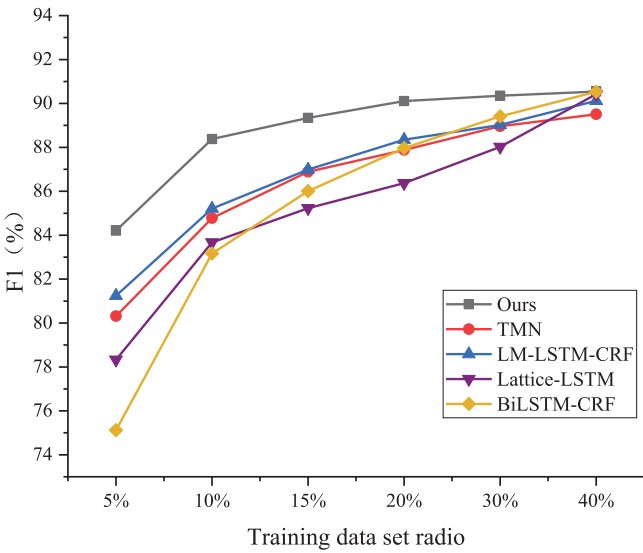

**Figure 4 Performance comparison.** As shown in Figure 4, the BiLSTM-CRF model exhibits the poorest performance among the five models, achieving less than 10% of the test set accuracy, despite using a significantly larger proportion of the training dataset.

(TMN) (*Lin et al., 2020*), a concept and trigger matching network framework for entity triggers. The model proposed in this article is trained using 5% to 20% of the training data. In order to facilitate the labeling of entity triggers, the author preprocessed the original training data set, so that only one entity in each sentence is labeled correctly, and other redundant entities are labeled as "O". One of the advantages of this approach is that it can avoid the overlap of triggers between different entities in a sentence, which is convenient for the model to code the data, but at the same time, it may bring the disadvantage of doubling the amount of training data and increasing the training time. However, because this study is based on very small data sets, the author believe that the increased training time can be ignored. Therefore, in accordance with the annotation rules described above, the author annotate the entity trigger corresponding to a single entity in each training sentence. The F1-scores for the model are computed on the validation and test sets of the *corpus*. After that, the neural network algorithm is trained to derive the accuracy of sentiment view monitoring, represented by a confusion matrix in the binary classification process. To ensure the validity of the data, each experiment is repeated five times on the respective datasets.

## Comparison results

As shown in Fig. 4, the BiLSTM-CRF model exhibits the poorest performance among the five models, achieving less than 10% of the test set accuracy despite using a significantly larger proportion of the training dataset. On the ResumeNER dataset, the BiLSTM-CRF model demonstrates an F1 value of 75.33 when trained using only 5% of the training dataset, which is much lower compared to the LM-LSTM-CRF model, the Lite-LSTM model, the TMN model, and the model proposed in this article under the same training

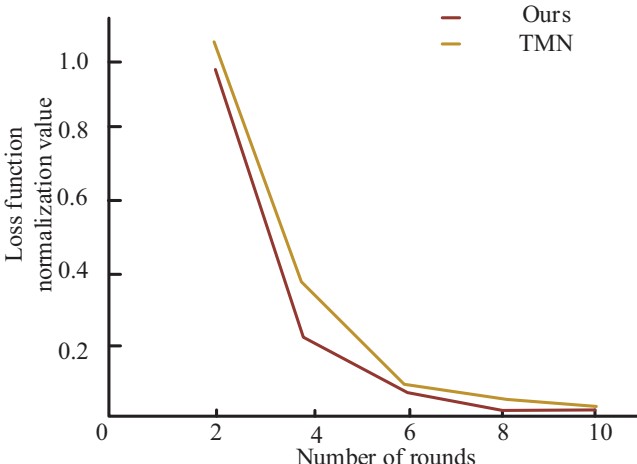

**Figure 5 Comparison of loss function convergence with TMN model.** To further demonstrate the superiority of my model over TMN in Chinese NER, the loss function curves of both models are presented.

conditions. Additionally, the performance of the BiLSTM-CRF model is the weakest when trained using 20% of the training dataset.

The F1 value of the BiLSTM-CRF model is 87.67, which is slightly higher than that of the TMN model. However, there is still a 3% difference in the F1 value between the BiLSTM-CRF and my proposed model. When the proportion of the training set is up to 40%, the F1 values of the Lite-LSTM and BiLSTM-CRF models tend to be equal to that of my proposed model. The performance variation of the TMN model is similar to that of my solution, with changes in the training set proportion, but the specific F1 value differs by approximately 3.8%. My proposed Chinese entity trigger and trigger matching network demonstrates good performance with very little data and is an effective labeling method and model.

To further demonstrate the superiority of my model over TMN in Chinese NER, the author presents the loss function curves of both models in Fig. 5. As shown in the figure, my model exhibits significantly faster convergence of the loss function after several training iterations using the same training dataset and achieves a smaller loss function value upon completion of training. Notably, the decreasing rate of my model's loss function is markedly higher than that of the TMN model in the initial few training rounds, and the value of my model's loss function converges to a lesser extent than that of the latter after several training iterations. These findings establish that my model outperforms the TMN model in the Chinese NER task. Additionally, the superior performance of my model not only translates to an improved evaluation metric F1 but also leads to reduced training time.

In addition, the author conducted a loss function effect experiment in the Resume NER dataset. The change in the classification accuracy of the trigger is obtained by adjusting the weight of the two loss functions. As shown in Table 2, I conducted experiments on 5%, 10%, 50%, and all data set samples. When the weight coefficient a is 0.3, the classification accuracy of the trigger is 0.977. When the weight coefficient a is 0, it means that no loss
**Table 2  Trigger classification progression accuracy under different loss function weight coefficients.**

| a | 5% | 10% | 50% | 100% | Ave |
|---|---|---|---|---|---|
| 0 | 0.961 | 0.961 | 0.981 | 0.981 | 0.966 |
| 0.1 | 0.974 | 0.977 | 0.977 | 0.981 | 0.975 |
| 0.2 | 0.951 | 0.979 | 0.977 | 0.982 | 0.977 |
| 0.3 | 0.963 | 0.984 | 0.977 | 0.982 | 0.977 |
| 0.4 | 0.919 | 0.965 | 0.972 | 0.981 | 0.959 |
| 0.5 | 0.939 | 0.966 | 0.975 | 0.972 | 0.963 |
| 0.6 | 0.911 | 0.936 | 0.952 | 0.979 | 0.945 |
| 0.7 | 0.898 | 0.936 | 0.938 | 0.974 | 0.937 |
| 0.8 | 0.860 | 0.877 | 0.935 | 0.969 | 0.910 |
| 0.9 | 0.758 | 0.832 | 0.889 | 0.941 | 0.855 |
| 1 | 0.491 | 0.590 | 0.584 | 0.572 | 0.559 |

**Table 3  The accuracy of emotional opinion monitoring algorithm under different $n$ values.**

| $n$ value | 50 | 100 | 150 | 200 | 250 | 300 | 350 | 400 |
|---|---|---|---|---|---|---|---|---|
| Accuracy (%) | 1 | 0.9822 | / | 0.9756 | / | 0.9689 | / | 0.9517 |

function $L_2$ is added to the model, that is, the original model. As can be seen from Table 2, the average accuracy of the original model is 0.966. When the weight coefficient a is 1, the classification accuracy of the trigger is 0.559.

The present study evaluates the effectiveness of a sentiment topic monitoring algorithm in news. As other existing schemes only recognize named entities and do not detect sentiment views based on them, the author only assesses the performance of the model proposed in this article. As the training set is incomplete, recall cannot be calculated, and precision is chosen as the evaluation criterion. Table 3 presents the obtained results.

In this experiment, the identified company and place names from the ResumeNER dataset are selected as the detection objects, and the top most relevant web news is extracted as the sentiment topic results based on the ResumeNER dataset. Notably, the accuracy of topic detection is 100% when the first 50 most relevant data are selected. In comparison, the algorithm's accuracy slightly decreases as the value of n increases but remains above 95% in all cases.

The author compared and analyzed the true and predicted results to examine the relationship between true and predicted values in the classification of affective views. The resulting ROC graph is depicted in Fig. 6. Combined with the confusion matrix of classification results shown in Fig. 7, my model demonstrates a superior classification effect for actual and predicted labels. Specifically, the correct classification rate into Angry, Disgust, Fear, Surprise, and Neutral categories are 79%, 89%, 75%, 94%, and 91%, respectively. Each category's lowest classification error rate is 1%, and only 12% of Fear labels are classified as Angry.

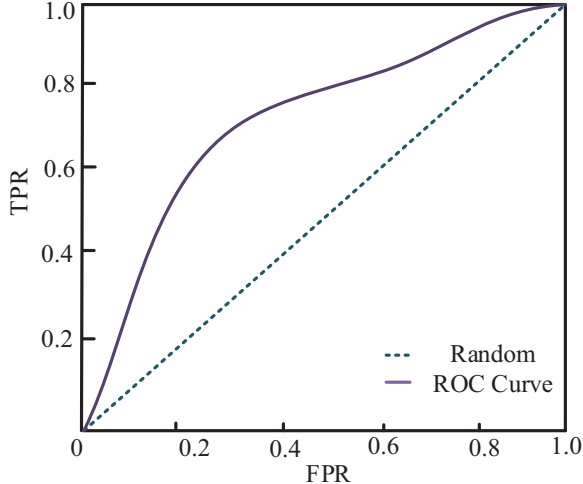

**Figure 6  ROC curve of my model.**               

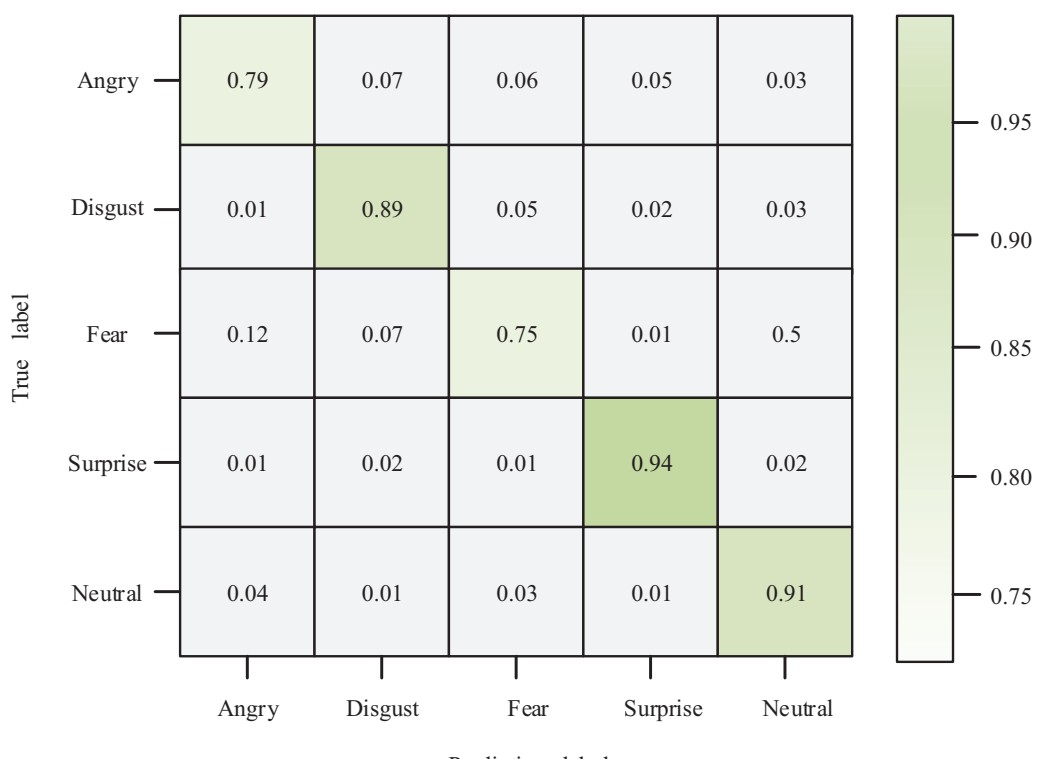

**Figure 7  Confusion matrix.**               

## CONCLUSION

This article addresses the analysis of online news data in the context of Internet development. To achieve this, the author proposes an efficient Chinese NER technology for a small amount of annotated data, starting with a method to annotate Chinese entity triggers automatically. An efficient NER model is trained by jointly training the sentence

vector and trigger vector through the trigger matching network and subsequently using the trigger vector as the attention query for the sequence annotation model. This approach provides a data basis for subsequent research on efficient NER technology while requiring no additional labor for additional supervision of the training dataset. The author further implements a neural network model for word vector calculation based on NER to tackle the increasing amount of online news information and the pressing demand for online sentiment monitoring. An algorithm for new word discovery and sentiment view monitoring is constructed. The author then constructs a recursive neural network model for online news sentiment opinion analysis by ignoring the contextual role of sentiment analysis. The experimental results demonstrate that the classification accuracy of triggers with different loss function weight coefficients achieved good performance. When classifying multiple emotional views, the classification error rates of Angry, Disgust, Fear, Surprise and Neutral were 1%. Cross-language domain research is one of the hot spots in the field of named entity recognition. At present, this article has made some achievements in the task of low resource named entity recognition in Chinese text field. How to further expand to English and other language fields will be a major challenge for the future research of this article.

### Funding
The authors received no funding for this work.

### Competing Interests
The authors declare that they have no competing interests.

### Author Contributions
- Manzi Tu conceived and designed the experiments, performed the experiments, analyzed the data, performed the computation work, prepared figures and/or tables, authored or reviewed drafts of the article, and approved the final draft.

### Data Availability
The code files are available in the Supplemental File.

The dataset is available at Zenodo:

Zhang. (2021). Sogou News *Corpus* (SOGOU) (Version v1) [Data set]. Zenodo. https://doi.org/10.5281/zenodo.5259056.

### Supplemental Information
Supplemental information for this article can be found online at http://dx.doi.org/10.7717/peerj-cs.1715#supplemental-information.

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
