# Peer review of "Named entity recognition and emotional viewpoint monitoring in online news using artificial intelligence"

_PeerJ Computer Science, doi:10.7717/peerj-cs.1715_

## Round 0.1 · original submission · Major Revisions

Based on the reviewers’ comments, you may resubmit the revised manuscript for further consideration. Please consider the reviewers’ comments carefully and submit a list of responses to the comments along with the revised manuscript.

Reviewer 1 ·

Basic reporting

1- There are too few keywords to summarize the content of the abstract;
2- Give more details about your model (sub-section 3.1)
3- The introduction of the pre-processing process of the collected data is lacking in the experiment, and the author can add it appropriately;
4- Enhance the quality of figures 1, 2 and 3. They are not very clear
5- Prived a colored Figure 8
6- Figure 8 in the experiment only has the confusion matrix of the method in this paper, I would suggest the author to add the confusion matrix of other models for comparison;
7. The conclusion is too short and insufficient to summarize the content of the whole paper.

Experimental design

.

Validity of the findings

.

Additional comments

no comment

Reviewer 2 ·

Basic reporting

as in the comments section

Experimental design

as in the comments section

Validity of the findings

as in the comments section

Additional comments

This paper discusses the analysis of network news data in the context of the development of the Internet. In order to achieve this, the author proposes an efficient Chinese NER technique for a small amount of annotated data. The sentence vector and the trigger vector are trained jointly through the trigger matching network, and then the trigger vector 365 is used as the attention query of the serial annotated model. To train an effective NER model. Superior performance is achieved using only 20% proportional training datasets compared to the 40% proportional training datasets required for traditional models. However, the author needs to modify the following points to improve the article:
1) There are too many keywords, the author needs to re-select according to the content of the abstract;
2) The author should add more background in the introduction section, and the description of the model should be placed in the related works section;
3) In section 3.3, how is it possible to reduce ambiguity by using multiple vocabulary representations?
4) The data processing in section 4.1 is not clear, but only describes the evaluation model and some network frameworks;
5) Compared with TMN model, how can the training time be reduced?
6) The ROC curve in Figure 7 and the confusion matrix in Figure 8 both lack relevant introduction;
7) The conclusion section only summarizes the general content of this paper, and lacks the prospect of the future;
8) The language standardization of the article needs to be strengthened.

Reviewer 3 ·

Basic reporting

In recent years, network news has become an important source for Internet users to obtain social information. However, the overwhelming news content will prevent users from obtaining key information. This paper uses deep learning technology to integrate NER into the artificial environment, introducing an automatic annotation method for Chinese entity triggers and an NER model. This model can achieve high precision with a small number of training data sets, and compared with the comparison model, the proposed model has better accuracy and faster convergence rate, but it still has the following shortcomings:
1. The first half of the abstract describes the logical confusion and suggests the author to write according to the background and the smoothness of the method;
2. What exactly does the first paragraph of the introduction refer to in terms of obtaining effective information from large amounts of data which cannot be vague;

Experimental design

3. The related works section should add a description of technologies related to online news;
4. What are the functions of named entities and the monitoring of emotional views, and what are the core innovations?
5. The system parameters of the experiment can be summarized in a table by the author;
6. The author needs to introduce the comparison model used in the experiment in detail;

Validity of the findings

7. When the weight coefficient is 0 in the experiment, the accuracy is higher, and the author needs to explain whether the weight coefficient is useless;
8. The structure of each chapter of the article should be the same, and I would recommend that the author add a paragraph at the beginning of each chapter to explain the chapter;
9. The references need to select some excellent articles in the past two years and keep the format consistent.

---

## Round 0.2 · accepted · Accept

Congratulations, the reviewers are satisfied with the revised version of the manuscript and have recommended the acceptance decision.

Reviewer 1 ·

Basic reporting

no comment

Experimental design

Author has done required changes

Validity of the findings

Author has done required changes

Reviewer 2 ·

Basic reporting

as in comments section

Experimental design

as in comments section

Validity of the findings

as in comments section

Additional comments

The manuscript is improved as suggested and commented to the author. I have no further concerns and would like to accept the paper for publication.

Reviewer 3 ·

Basic reporting

The article seems to be improved with well defined updates

Experimental design

The comments are well answered with clear explanation.

Validity of the findings

it seems to be a fair display, and justification of findings after the updates, can be accepted